# Active Crack Obstruction Mechanisms in Crofer^®^ 22H at 650 °C

**DOI:** 10.3390/ma15186280

**Published:** 2022-09-09

**Authors:** Torsten Fischer, Bernd Kuhn

**Affiliations:** Institute of Energy and Climate Research (IEK), Microstructure and Properties of Materials (IEK-2), Forschungszentrum Jülich GmbH, 52425 Jülich, Germany

**Keywords:** fatigue, crack propagation, active crack obstruction mechanisms, frequency, ferritic steels, Laves phase

## Abstract

Increased cyclic loading of components and materials in future thermal energy conversion systems necessitates novel materials of increased fatigue resistance. The widely used 9–12% Cr steels were developed for high creep strength and thus base load application at temperatures below 620 °C. At higher temperature, these materials present unstable grain structure, prone to polygonization under thermomechanical fatigue loading and limited resistance to steam oxidation. This seminal study compares thermomechanical fatigue resistance and long crack propagation of the advanced ferritic-martensitic steel grade 92 and Crofer^®^ 22H, a fully ferritic, high chromium (22 wt. %) stainless steel, strengthened by Laves phase precipitation. Crofer^®^ 22H features increased resistance to fatigue and steam oxidation resistance up to 650 °C. Both thermomechanical fatigue (crack initiation) and residual (crack propagation) lifetime of Crofer^®^ 22H exceeded that of grade 92. The main mechanisms for improved performance of Crofer^®^ 22H were increased stability of grain structure and “dynamic precipitation strengthening” (DPS). DPS, i.e., thermomechanically triggered precipitation of Laves phase particles and crack deflection at Laves phase-covered sub-grain boundaries, formed in front of crack tips, actively obstructed crack propagation in Crofer^®^ 22H. In addition, it is hypothesized that local strengthening may occur near the crack tip because of grain refinement, which in turn may be impacted by testing frequency.

## 1. Introduction

Future thermal energy conversion systems such as concentrating solar power (CSP) [1,2] and thermal energy storage (TES) based on molten salts as exemplary thermal energy generation and storage technologies and downstream Clausius-Rankine steam processes exemplary for electricity generation in solar thermal or heat storage power plants [3], etc. will be characterized by highly dynamic changes in operating conditions. Besides low-frequency operating cycles such as the start-up and shut-down processes in heat storage power plants, medium-frequency load changes are increasingly to be expected, for example, in concentrating solar receivers and downstream heat storage units as a result of cloud passage. This results in pronounced thermomechanical fatigue loading of the structural materials applied. In general, cyclic loading causes significant damage earlier than pure creep loading does [4,5,6]. The advanced ferritic-martensitic (AFM) 9–12% Cr steels, widely utilized in energy conversion, were developed primarily for high creep strength and application temperatures of 580–620 °C [7,8]. Application of the 9% Cr steels beyond 620 °C is not possible due to limited steam oxidation resistance [7]. This results in the need for new structural materials with increased thermomechanical fatigue (TMF) resistance and higher temperature capability for future thermal energy conversion systems.

A potential candidate material to base further development on is Crofer^®^ 22H. Crofer^®^ 22H is a high chromium fully ferritic steel (20–24 wt. % Cr [9]), strengthened by a combination of solid solution hardening and intermetallic (Fe,Cr,Si)_2_(Nb,W) Laves phase particle precipitation [10]. It was developed by the Institute for Microstructure and Properties of Materials (IEK-2) in cooperation with VDM Metals GmbH for the application in high temperature solid oxide fuel cell interconnectors. Because of its high chromium content, Crofer^®^ 22H offers adequate steam oxidation resistance up to 650 °C [10]; moreover, some of the trial alloys from Crofer^®^ 22H development exhibited excellent creep behaviour in the temperature range of 600–650 °C [10]. With fatigue resistance gaining in importance, creep strength still remains significant, because it results in low component wall thickness and for this reason helps in diminishing thermal stresses during operation. In addition, Crofer^®^ 22H showed high resistance to fatigue loading [11,12]. Ferritic “Crofer^®^ 22H-type”, high chromium, Laves phase strengthened steels demonstrated quite stable grain structure under TMF loading, while the martensite lath structure of the 9–12% Cr AFM steels proved unstable and prone to polygonization [13]. Polygonization is regarded as a decisive reason for the shorter technical lifetime of 9–12% Cr steels [14]. In Crofer^®^ 22H-type steels, thermomechanical loading, causing a quasi-permanent strengthening effect by thermomechanically triggered Laves phase particle generation [15], is considered as a main reason for superior fatigue resistance. Moreover, cyclic or plastic deformation results in accelerated Laves phase precipitation and particle refinement [15].

Total material lifetime consists of technical lifetime until cracking (short crack initiation and growth, characterised by low cycle fatigue (LCF) or TMF lifetime) and residual lifetime (long crack growth, characterised by crack propagation rate). The residual lifetime potential of a 9–12% Cr steel was examined in air [14,16] as well as steam [4,17,18] and showed an attractive possibility to extend the lifetime of a component considerably without significant additional costs. However, until recently, there was no draft directive for the application by technical surveillance authorities. This could change with the development of an advanced draft directive by TÜV Nord EnSys GmbH & Co. KG, Hamburg, Germany [19].

Proper utilization of technical and residual lifetime by improved codes in combination with materials of improved fatigue resistance may open up new possibilities in future design against fatigue. For these reasons, the focus of the present study was on long crack propagation behaviour and the identification of active crack obstruction mechanisms in Crofer^®^ 22H at an operating temperature of 650 °C in direct comparison with AFM grade 92 steel. For this purpose, extensive crack growth experiments in combination with scanning and transmission electron microscopy investigations were performed.

## 2. Materials and Methods

### 2.1. Experimental Materials

Ferritic-martensitic T/P 92 is a 9Cr-1Mo steel, which was developed to achieve higher permissible creep stress than T/P 91 [20]. For this purpose, T/P 92—in comparison to T/P 91—features a decreased Mo content of 0.5% and an addition of 1.8% of W [20]. The typical microstructure of a normalized and tempered martensite, consisting of prior austenite grain boundaries (PAGBs), packet and lath boundaries is depicted in Figure 1 [21]. Solid solution (Cr, W, V, Nb), precipitation (MX (M: V, Nb; X: N and M_23_C_6_ (M: Mo, Cr)) and sub-grain strengthening are employed in 9Cr-1Mo steels. However, during long-term creep, secondary, intermetallic Laves and Z-phase [22] precipitates appear. The M_23_C_6_ particle size is usually much smaller in the grain interiors than at the PAGBs [21]. Moreover, the size of M_23_C_6_ particles is much larger than the size of MX carbonitrides [21]. Typically the MX carbonitrides are predominantly distributed within the lath matrix, while M_23_C_6_ precipitates primarily along PAGBs and packet, block and lath boundaries (Figure 1) [21].

Grade 92 steel specimens (in normalized and tempered state) for the crack growth threshold experiments of this study were taken from a forged bar supplied by BGH Edelstahl Freital GmbH, Germany. The chemical composition of the material is displayed in Table 1.

The second steel investigated in the presented work was Crofer^®^ 22H. The formation of a duplex Cr_2_O_3_/(Mn,Cr)_3_O_4_ surface oxide scale results in excellent oxidation resistance [10] at high temperature. In contrast to the 9Cr-1Mo steels, fully ferritic, high chromium stainless steels such as Crofer^®^ 22H cannot be strengthened by carbide and nitride precipitation because of the limited solubility of C and N in the ferrite matrix. For this reason, Crofer^®^ 22H is strengthened by a combination of solid solution and intermetallic (Fe,Si,Cr)_2_(Nb,W) Laves phase precipitate hardening [23]. Nb is a strong carbonitride former. To make the best use of Laves phase precipitation, the C and N contents have to be kept to a minimum (<0.01 wt. %). Even low levels are sufficient to consume Nb by formation of primary Nb/Ti-MX (with X = C, N [24]) and by this, significantly reduce the amount of Nb available for precipitation of fine Laves phase [25,26,27,28] precipitates in the surrounding alloy matrix (Figure 2a, reproduced from [29]). In this type of Laves phase strengthened steel, TMF [11,12,13] and long crack growth behaviour [15] are significantly impacted by additional, thermomechanically induced precipitation of Laves phase particles. The characteristic microstructural features of recrystallised and precipitation-annealed (RX + PA: 1075 °C/22 min/air cooling + 650 °C/2 h/water quenching) Crofer^®^ 22H are fine, evenly distributed (except in the neighbourhoods of primary MX-particles) intragranular Laves phase particles and Laves phase-covered high-angle grain boundaries (HAGBs) with adjacent particle-free zones (PFZ, Figure 2b).

Crofer^®^ 22H material from VDM Metals, Germany was TMF- and FCG/threshold-tested in the recrystallized (1050 °C/5 + minutes, depending on cross-section thickness/rapid air cooling) and in the recrystallized and subsequently precipitation-annealed (PA) state (2 h/650 °C/water quenched). Samples from three individual Crofer^®^ 22H rolled sheets were taken from one melt and tested to elucidate the impact of material batch scatter on short- and long-crack propagation. The chemical compositions were analysed by inductively coupled plasma optical emission spectrometry and are listed in Table 1.

### 2.2. Thermomechanical Fatigue Experiments

The total strain controlled out-of-phase (oop) TMF tests in a temperature range from 50 to 650 °C were conducted according to the European Code-of-Practice [30] with cylindrical specimens with a gauge length of 15 mm and a diameter of 7 mm, applying servo-hydraulic fatigue testing machines equipped with induction heaters. The temperature was monitored with type R sling thermocouples, and the specimen strain was recorded directly at the gauge length utilising high-temperature extensometers. The heating rate to maximum temperature and the cooling rate to minimum temperature were dT/dt = 10 Ks^−1^, whereby controlled cooling was accomplished by compressed air. The amount of cooling air was actively controlled to match the scheduled cooling rate by application of an electromechanical mass flow controller. In order to keep creep to a minimum, no holding times were integrated into the cycle. Below 200 °C, cooling was retarded due to an insufficient amount of cooling air, resulting in a cooling half-cycle of 85 s. The resulting TMF cycle is schematically depicted in Figure 3. All TMF experiments were conducted in the low-cycle regime, because the focus of the study was to investigate crack initiation and propagation mechanisms.

### 2.3. Fatigue Crack Growth Experiments

Fatigue crack growth (FCG) testing was performed in a servo hydraulic Instron (Norwood, MA, USA) Model 1343 testing machine equipped with an inductive heating system. Due to limited material availability, compact tension (CT) specimens with dimensions of width (W) = 40 mm, thickness (B) = 10 mm and a machined notch depth of a_n_ = 10 mm were utilized. The specified dimensions were in accordance with the ASTM E647-11 fatigue crack growth testing standard [31]. The CT specimens were pre-cracked to a starter crack length a_0_ to width ratio (a/W) of 0.4 by cyclic loading at ambient temperature in an Instron Model 1603 resonance testing machine. The direct current potential drop (PD) technique was utilised to continuously measure the crack length in the FCG tests. All experiments were conducted with a sinusoidal waveform until the termination criterion of a/W = 0.7 was reached, with the exception of the Crofer^®^ 22H, PA, 20 Hz FCG experiment, which—in the range of low stress intensities with decreasing da/dN by increasing ΔK —was terminated for microstructure investigations (cf. Table 2).

In addition, all tests were carried out at a constant load ratio R of 0.1 at 650 °C. The frequencies (5 Hz and 20 Hz) and the atmosphere (lab air and vacuum) were varied. A detailed test matrix is given in Table 2. In order to carry out experiments in vacuum, the servo-hydraulic testing machine (Instron Model 1343, Norwood, MA, USA) was additionally equipped with a Leybold Heraeus vacuum chamber (chamber volume 270 L, vacuum 4 × 10^−5^ mbar, and a leak rate of 1 × 10^−3^ mbar L/s).

The cyclic stress intensity factor ΔK for the CT specimen was determined as described in ASTM E647-11 [31]. The calculation of the cyclic crack growth rate was accomplished by a 7-point polynomial method according to ASTM E647-11 [31]. The threshold values (ΔK_th._) were determined by the stepped force shedding method specified in ASTM E647-11 [31].

### 2.4. Microstructure Investigation

From selected experiments (Crofer^®^ 22H), the fracture surfaces were cut from the CT specimens and measured, excluding the pre-cracked and residual fracture surface regions, in order to investigate the microstructure at defined regions/positions of the cyclic crack growth curve. Subsequently, longitudinal sections were prepared for microscopic analysis. For this purpose, the fracture surfaces were hot embedded (with the exception of FCG tests), terminated in the “kink” regions (i.e., da/dN↓ with ΔK↑) for metallographic preparation, ground and polished. For high resolution microstructure investigation, a Zeiss Merlin (Oberkochen, Germany) field emission scanning electron microscope (FESEM) was used.

The specimens for electron backscatter diffraction (EBSD) analyses were additionally polished for 2–3 h to a sub-micron finish in colloidal silica suspension. EBSD investigation was performed on a Zeiss Merlin SEM (Oberkochen, Germany), equipped with an AZtec, Oxford Instruments (High Wycombe, UK); NORD LYS 2 camera. Schmid factors were determined using HKL Channel 5 software (version number: 5.12.74.0) from Oxford Instruments (High Wycombe, UK).

The specimens from FCG experiments terminated in the “kink” region (i.e., da/dN↓ with ΔK↑) were cold embedded in epoxy resin under vacuum for detailed examination of the region in the vicinity of the crack tip. Detailed analysis of the front of crack tips was carried out by a Zeiss Libra 200 (Oberkochen, Germany) transmission electron microscope (TEM) with an acceleration voltage of 200 kV. The images were taken in scanning transmission electron microscope (STEM) mode. Lamellae with a thickness 100–150 nm at a size of 10 × 10 μm^2^ were cut in front of the crack tip using gallium ions in a Zeiss Auriga Focused Ion Beam (FIB) device (Oberkochen, Germany). 

For microstructure analysis of grade 92, the specimen was electrolytically etched in 10% oxalic acid at a voltage of 2 V for 10 s.

## 3. Results and Discussion

### 3.1. Total Lifetime Potential of Crofer^®^ 22H

In general, the total lifetime of a material is classified into the range up to the technical crack and the consecutive residual lifetime. The thermomechanical fatigue (TMF) lifetime of Crofer^®^ 22H until technical cracking was approx. 40% higher than that of grade 92 steel (Figure 4a) in the application-relevant temperature range up to 650 °C. At 650 °C, the residual lifetime of Crofer^®^ 22H, which is largely determined by the crack propagation rate, was up to 5 times longer (Figure 4b).

Crofer^®^ 22H exhibited a very stable grain structure under TMF loading (Figure 5a), while the martensite lath structure of the 9–12% Cr steels proved to be unstable, with polygonization occurring (Figure 5b).

The phenomenon of polygonization was an important reason for the reduced total lifetime and especially the comparatively rapid failure after crack initiation of 9–12% Cr steels compared to ferritic steels [13,14]. The focus of this study was to identify the mechanisms responsible for the retarded crack propagation and increased residual lifetime of Crofer^®^ 22H.

### 3.2. Residual Lifetime Potential—Threshold Behaviour

Besides the stable crack growth region, the threshold behavior is of considerable importance for the evaluation of the residual lifetime. Figure 6 shows the results of the threshold experiments carried out with precipitation-annealed Crofer^®^ 22H (PA) in comparison to grade 92 steel at 5 and 20 Hz at 650 °C. In 9–12 Cr steels, an increase in threshold with decreasing frequency was observed [5,16]. The same was encountered with the grade 92 material in this study. In contrast, the threshold value of Crofer^®^ 22H (PA) increased with increasing frequency, resulting in a drawback of 0.8 MPa√m at 5 Hz, but an advantage of 1 MPa√m at 20 Hz (cf. Table 3) over grade 92. A possible explanation for this “inverse” dependency would be pronounced plasticization at increased frequency (20 Hz). At high frequency, dislocations have less time to overcome obstacles; additionally, more precipitates form because of stronger plasticization [32,33]. Therefore, it is likely that in the 20 Hz experiment, the crack front did not completely pass through the plastic zone in front of the crack tip because of strong plasticization and increased precipitation, leading to an overestimation of the threshold.

With regard to the significance of the threshold for the residual lifetime potential of an optimized material based on Crofer22^®^ H, an extensive quantitative microstructure evaluation of crack tips near regions to qualify this working hypothesis should be tackled in the future.

### 3.3. Residual Lifetime Potential—Stable Crack Growth Behavior

#### 3.3.1. Impact of Precipitation Annealing

In Section 3.1, the enhanced resistance of Crofer^®^ 22H (in the recrystallized state (RX)) against short crack propagation was demonstrated. In the following, the impact of precipitation annealing (PA) on the propagation of long cracks will be outlined.

In experiments started in the precipitation-annealed (PA) state, the precipitation of additional (Fe,Cr,Si)_2_(Nb,W)-Laves phase particles leads to an initial drop in crack propagation rate (Figure 7) despite increasing stress intensity ΔK (at f = 20 Hz, 650 °C), and consequently, to a visible “kink” in the crack propagation curve at low stress intensities.

This “kink” (i.e., the range of decreasing crack propagation rate) accounts for about half of the cycles to failure (~1 million cycles). After the curve minimum, plateaus of approximately constant crack growth rate with increasing ΔK were encountered (highlighted by a green box in Figure 7) in the crack propagation curve. Both the kinks (exemplified in Figure 7) and the plateaus in cracking velocity were repeatedly observed in precipitation-annealed Crofer^®^ 22H (PA), too. Figure 8 shows local minima (plateaus) in both the RX and PA states in da/dN vs. ΔK and ΔK vs. time plots. In order to enable direct comparison of the PA and RX states, the time in the PA state was corrected by adding a heat treatment duration of 1 h (at 650 °C) to the experimental time. In the RX state, it took about 4.4 h to initiate plateau formation (“initial” range, Figure 8), while it took about 5.9 h in the PA state (“initial” range measured from the global minimum (kink) in the PA state, Figure 8). The global minimum was chosen as the starting point for the initial range in the PA state, because from this point onwards, stress intensities, comparable to the RX state (shown in Figure 8 by dotted lines intersecting the *x*-axis), occurred. The stress intensity increases significantly faster with time in the RX than in the PA state. This indicates that under thermomechanical loading (I) Laves phase precipitation was accelerated [11,13] when starting from the RX state, and (II) additional Laves phase precipitation occurred after a prolonged (+1.5 h) incubation time when starting from the precipitation-annealed state. This implies certain reserves against crack propagation even in the PA state (rising branch, Figure 7), which is underlined by the fact that in both states, the number of bearable cycles ranges up to about 1 million.

Neither of the two initial material states presented a classical Paris region [34] in the respective crack propagation curves. Nevertheless, a classical Paris fit [34], strongly underestimating the real residual lifetime potential, was performed for comparative purposes (Figure 9a). Both the coefficient C and the exponent m were lower in precipitation-annealed state (cf. Table 4). Even extrapolation of the Paris fit line until intersection with the y-axis still led to significant underestimation. Utilization of a fifth-degree polynomial in contrast fit the cyclic crack growth curve well enough to provide full exploitation of the residual lifetime potential (Figure 9b).

#### 3.3.2. Active Crack Obstruction Mechanisms

To identify the mechanisms responsible for such unconventional cyclic crack growth curves, the microstructures were investigated in the initial state and after termination of experiments in “plateau” regions (Figure 7). In kink range (where cracking velocity decreased despite increasing ΔK), the formation of sub-grains was observed (Figure 10a). Sub-grain formation also occurred in the range of higher stress intensities (where cracking velocity increased with increasing ΔK), but led to plateau formation (Figure 10b). In this region, the crack path appeared cascade-like with fissured edges, while a straight crack path with sleek edges was observed before the transition into the “plateau” area. This implies crack deflection at sub-grain boundaries (SGBs) to be the root cause for the cascaded crack path and the fissured crack edges. Moreover, secondary cracks grew along SGBs.

Sub-grain formation initiated in front of the high-angle grain boundary (HAGb) and then continued into the adjacent grain (Figure 11). To analyse the causes of sub-grain formation, Schmid factors of the slip systems parallel to the applied force and the misorientation of the two neighbouring grains were determined. In the following discussion, a low number of screw dislocations was assumed, otherwise Schmid’s law would not be applicable. The crack in grain I propagated at an angle of 45° to the tensile load, i.e., along the path of highest shear stress. This was consistent with the determined Schmid factors of ~0.5 near the crack path. Furthermore, all slip systems in grain I were activated earlier, because of the higher Schmid factors evaluated (cf. Table 5). Consequently, shear stress acted on the slip systems. In addition, the matching Schmid factors of the ($\left\{110\right\}\;\langle \bar{1}11\rangle$) and $\left\{112\right\}\;\langle 11\bar{1}\rangle$) systems demonstrate that double slip had occurred. In grain II, the crack growth direction changed (Figure 11) near the grain boundary. The misorientation angle between grain I and II was measured to about 30°. According to the lower Schmid factors (cf. Table 5), the slip systems in grain II were activated later. During deformation, dislocations accumulate at grain boundaries. If a critical dislocation density is then reached, the total energy may be lowered by the rearrangement of dislocations into groups. Mutual reduction of the strain field of each dislocation in this case results in lowered total stored energy [35]. This indicates that certain orientation relationships favour sub-grain formation.

For a more detailed investigation of the microstructural mechanisms in precipitation-annealed Crofer 22H (PA), a fatigue crack growth (FCG) experiment at 650 °C and 20 Hz was terminated in the initial low stress intensity region. In this specimen, several crack fronts were formed (Figure 12a). An arrested crack (position I in Figure 12a) and the most advanced crack (position II in Figure 12a) were selected for further microstructure investigation. Sub-grain formation was observed in the case of the arrested crack (position I). It is noteworthy that sub-grain boundaries, formed parallel to the crack, obviously benefit crack deflection. In contrast, no sub-graining occurred at position II, where the experiment was terminated.

Transmission electron microscopic (TEM) investigations in front of the arrested crack (position I) unveiled the formation of a sub-grain boundary (Figure 12b and Figure 13). It is assumed that dislocation lines formed due to plastic deformation in front of the crack tip. These were then arrested at Laves phase particles, leading to the formation of SGBs and consequently resulting in grain refinement in front of the crack tip. Grain refinement causes an increase in the cyclic hardening potential [36], which could lead to enhanced crack growth resistance. The relationship between dislocation density and sub-grain size can be described by the following equation [37]:(1)$$d=\frac{K}{\sqrt{\varphi }}$$
with φ denoting dislocation density, *d* sub-grain size and *K* a material constant. This could indicate varying dislocation densities, caused by different loading conditions (e.g., testing frequencies), influencing the resulting sub-grain size and thus the local strengthening by grain refinement in front of the crack tip. For example, higher testing frequency could cause higher dislocation density and consequently smaller sub-grain size. Furthermore, the crack was deflected at sub-grain boundaries and at Laves phase particles (Figure 13), resulting in a prolongation of the crack path and consequently increased residual lifetime.

A SEM comparison of the crack tips clearly shows differing numbers and sizes of precipitates in the surrounding matrix (cf. Figure 12b vs. Figure 13). In TEM resolution, rod-like particles of differing sizes seem to have aligned according to grain orientation (cf. Figure 12c and Figure 13). In α-iron disc- (on {111} matrix planes) and lath-shaped (in the overaged condition by trend elongated in <112> matrix direction), hexagonal Fe_2_Nb Laves phase particles (similar to the mixed (Fe,Cr,Si)_2_(Nb,W)-Laves phase in this study) were documented in [38]. Furthermore, Ref. [38] stated a single orientation relationship between the Fe_2_Nb Laves phase and the α-matrix of ${\left\{11\bar{2}0\right\}}_{Fe_{2}}Nb∥{\left\{111\right\}}_{\alpha }\;:\;{\langle 0001\rangle }_{Fe_{2}}Nb\;∥{\langle 11\bar{2}\rangle }_{\alpha }$. Hence, dislocations may bypass the Laves phase particles more easily if the active slip system and the precipitates have similar orientations.

However, it should be noted that a possible 2D “cutting” effect, caused by specimen preparation, may have had an effect on these observations, and more detailed investigation is necessary to further elucidate the obvious differences in particle size distributions.

Even in the precipitation-annealed state, thermomechanically triggered precipitation of additional Laves phase particles (Figure 14) occurs at sufficient plastic deformation in Crofer^®^ 22H. These newly formed precipitates additionally obstruct crack growth. Compared to thermal loading only (Figure 14a), a significantly higher particle density was encountered under thermomechanical loading (Figure 14b). In addition, newly formed SGBs are rapidly covered by nucleating precipitates and act as further obstacles for dislocation movement and crack propagation.

#### 3.3.3. Impact of Frequency

Reduced testing frequency (from 20 Hz to 5 Hz) caused the crack propagation rate of precipitation-annealed (PA) Crofer^®^ 22H to increase (Figure 15, cf. coefficient C and exponent m in Table 6), which is a well-known behavior of steels [5,14]. At 5 Hz, the initial kink was absent. The working hypothesis to explain this behavior encompassed a combination of diminished strain hardening potential and more time for dislocations to overcome obstacles (e.g., Laves phase particles) at low frequency. However, at low frequency (5 Hz), higher stress intensity was required to initiate crack propagation (ΔK_Start, 5Hz, air_~11 MPa√m vs. ΔK_Start, 20Hz, air_ ~9 MPa√m; a detailed description of the determination of ΔK_Start_ is given in [17]). Furthermore, the ΔK_Start_ value of the 5 Hz experiment coincided with the ΔK value at the minimum of the 20 Hz experiment. In addition, the curve slope of the 5 Hz experiment after the kink (Figure 15) was comparable to the one encountered initially in the 20 Hz experiment.

Interestingly, the same absence of the initial drop of da/dN with rising ΔK along with the coinciding ΔK_Start_ value of 11 MPa√m was observed in recrystallized Crofer^®^ 22H (RX) at 20 Hz (Figure 7), too. This may indicate that the root cause of the absence of the initial drop is not testing frequency. Rather, a limit value stress intensity ΔK_LV_ (to avoid confusion with the threshold value ΔK_th._) may exist, from which the outlined strengthening mechanisms (i.e., thermomechanically induced precipitation and sub-grain formation, crack defection at sub-grain boundaries) cannot longer overweigh the increasing stress intensity. To verify the assumption further, 20 Hz FCG experiments were carried out at the same load ratio with recrystallized and precipitation-annealed material in vacuum. Vacuum was chosen as the testing environment to eliminate corrosive mechanisms and thus achieve higher ΔK_Start_ values at comparable load ratios.

For both material states, the ΔK_Start,_ values measured in vacuum were above the 11 MPa√m measured in air (PA, vac. ~14.7 MPa√m, RX, vac. ~13.5 MPa√m). Furthermore, “kink” regions of decreasing da/dN with increasing ΔK did not occur (Figure 16) in vacuum. In consequence, the limit value ΔK_LV_, from which the strengthening mechanisms cannot longer overweigh the increasing stress intensity, has to be approximately 11 MPa√m (dotted line in Figure 16). Because kink formation was not observed in air (5 Hz experiment, cf. Figure 16), corrosive effects can be excluded as the root cause for kink formation.

#### 3.3.4. Impact of Material Batch Scatter

The results presented so far were all gathered with specimens taken from a single Crofer^®^ 22H plate. To elucidate material homogeneity and batch scatter further, 20 Hz FCG experiments were carried out with specimens taken from two additional plates, but from the same alloy heat. Obviously, the kink phenomenon did not occur in specimens taken from the two additional sheets (II and III, Figure 17). However, initiation of crack propagation in both the additional sheets necessitated stress intensities beyond the evaluated ΔK_LV_ value, which explains the absence of the kink. On the other hand, plateau formation at high stress intensities occurred in specimens taken from all three sheets (cf. green boxes in Figure 17), which means the previously identified crack obstruction mechanisms were active despite the absence of the kink at low stress intensity. As the comparison of the FCG parameters (Table 7) demonstrates, the crack growth behavior in Crofer^®^ 22H in general presented significant scatter, which was clearly related to the Nb/Ti-MX (Figure 18a) induced inhomogeneity in the precipitation of fine, strengthening Laves phase particles (cf. Figure 2a and Figure 18b).

The alloy design of Crofer^®^ 22H for this reason has to be considered conservative in terms of utilization of the outlined active crack obstruction mechanisms. Avoiding MX-related microstructural inhomogeneity is promising for the development of improved high-performance ferritic steels with maximized utilization of active crack obstruction mechanisms [15] for superior fatigue resistance.

## 4. Conclusions

The total fatigue life potential of Crofer^®^ 22H-based steels was investigated in comparison to state-of-the-art ferritic-martensitic steel grade 92 for application in future thermal energy conversion systems. Application relevant, oop TMF tests were performed in the temperature range from 50 to 650 °C, and FCG experiments at 650 °C. Moreover, the proof temperature of 650 °C corresponded to the Laves phase precipitation temperature. The following conclusions can be drawn based on experimental data and the corresponding microstructural observations:-In Crofer^®^ 22H, PA, an inverse frequency dependency of the threshold value was observed. The exact reasons for this finding are not yet fully understood.-In Crofer^®^ 22H, RX, the crack propagation rate was up to half an order of magnitude lower compared to that of grade 92 steel.-No “classical” Paris regime occurred in Crofer^®^ 22H, PA at 20 Hz.-Instead of this, the formation of “kinks” (da/dN↓ at ΔK ↑) and local “plateaus” (da/dN = const. at ΔK ↑) was observed.-Thermomechanically induced Laves phase precipitation, crack deflection at Laves phase covered sub-grain boundaries (formed in front of crack tips) were identified as the responsible active crack obstruction mechanisms that caused these characteristic curve properties.-In addition, it is assumed that local strengthening occurred near the crack tip due to local grain refinement.-Testing frequency probably has an impact on the resulting sub-grain size and consequently on local strengthening by grain refinement (d = K/$\sqrt{\varphi }$ [37]).-The apparent “frequency effect” in air may not necessarily be related to frequency alone. In contrast, a ΔK_LV_ value may exist, above which the strengthening mechanisms can no longer counteract crack propagation. This ΔK_LV_ value may be exceeded in 5 Hz testing (already in the initiation of crack propagation), and consequently, kink formation will not occur.-Consumption of Nb from the matrix by coarse Nb/Ti-MX and subsequent inhomogeneity in the precipitation of fine, strengthening Laves phase particles were identified to cause significant batch variation in the crack propagation behavior in Crofer^®^ 22H.-Nevertheless, improved Laves phase strengthened stainless high performance ferritic (HiperFer) steels (similarly to Crofer^®^ 22H as a base for the development of improved grades) pose high potential for design against fatigue at high temperature.

## Figures and Tables

**Figure 1 materials-15-06280-f001:**
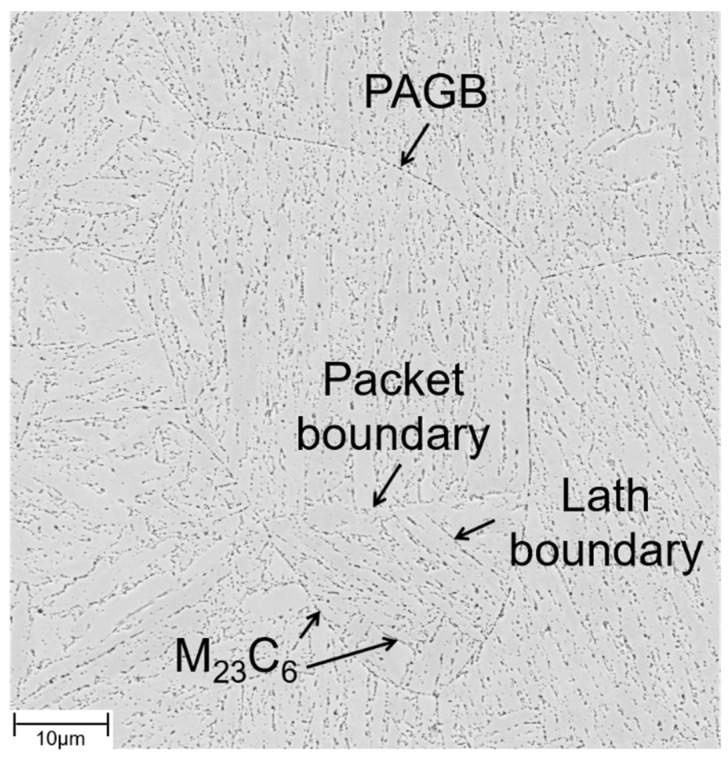
Characteristic microstructure of normalized and tempered grade 92 steel (SEM image).

**Figure 2 materials-15-06280-f002:**
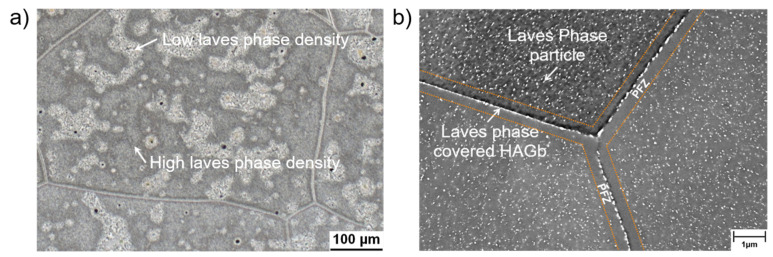
Characteristic microstructure of recrystallized and precipitation-annealed (RX + PA) fully ferritic Crofer^®^22 H steel. (**a**) Light optical resolution: Inhomogeneous Laves phase precipitate density (light areas: low density; dark areas: high density), caused by primary Ti/Nb-MX precipitates (reproduced from [30]). (**b**) SEM resolution: Particle-free zones (PFZ) located at high-angle grain boundaries (HAGB, reproduced from [11]).

**Figure 3 materials-15-06280-f003:**
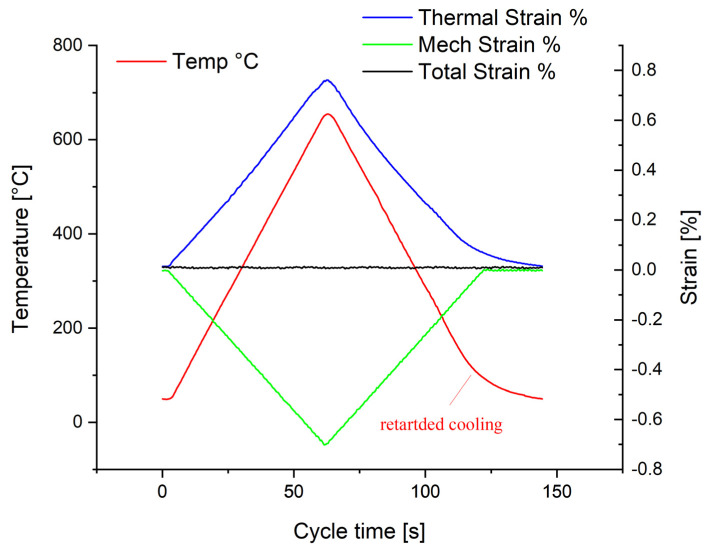
Schematic representation of an out-of-phase thermomechanical fatigue cycle with full (100%, i.e., ε_mechanical_ = −ε_thermal_) obstruction of thermal expansion (ΔT = 50–650 °C, dT/dt = 10 K/s).

**Figure 4 materials-15-06280-f004:**
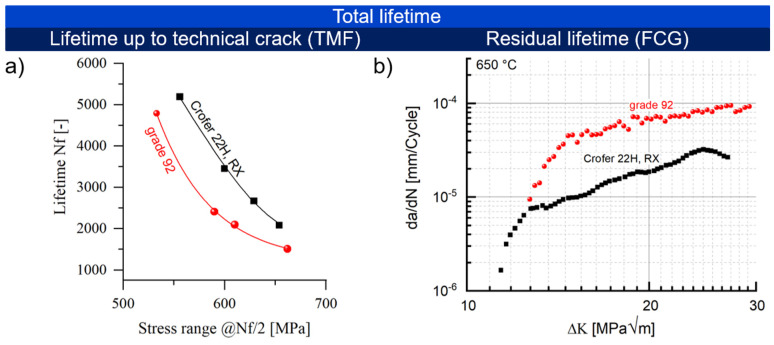
Total lifetime consisting of (**a**) the technical lifetime to cracking (characterized by (out of phase)-TMF lifetime (ε_mech_ = −ε_th_., ΔT = 50−650 °C, dT/dt = 10 Ks^−^^1^, th = 0 at T_min_ and T_max_ [15]) and (**b**) residual lifetime (characterized by the crack propagation rate; 650 °C, R = 0.1 f = 20 Hz) of Crofer^®^ 22H, RX and grade 92 [15].

**Figure 5 materials-15-06280-f005:**
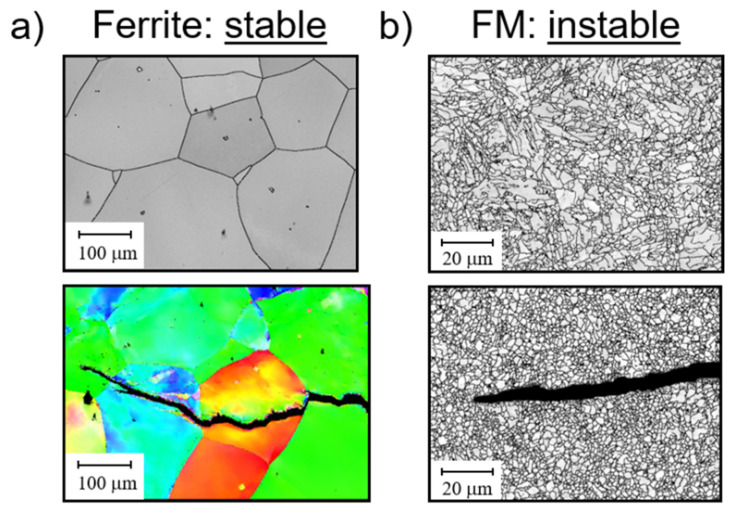
Comparison of grain structure in the initial state and in the vicinity of cracks induced by TMF loading in (**a**) a microstructurally stable ferritic (EBSD-image) and (**b**) an unstable ferritic-martensitic P91 [13] steel (SEM image).

**Figure 6 materials-15-06280-f006:**
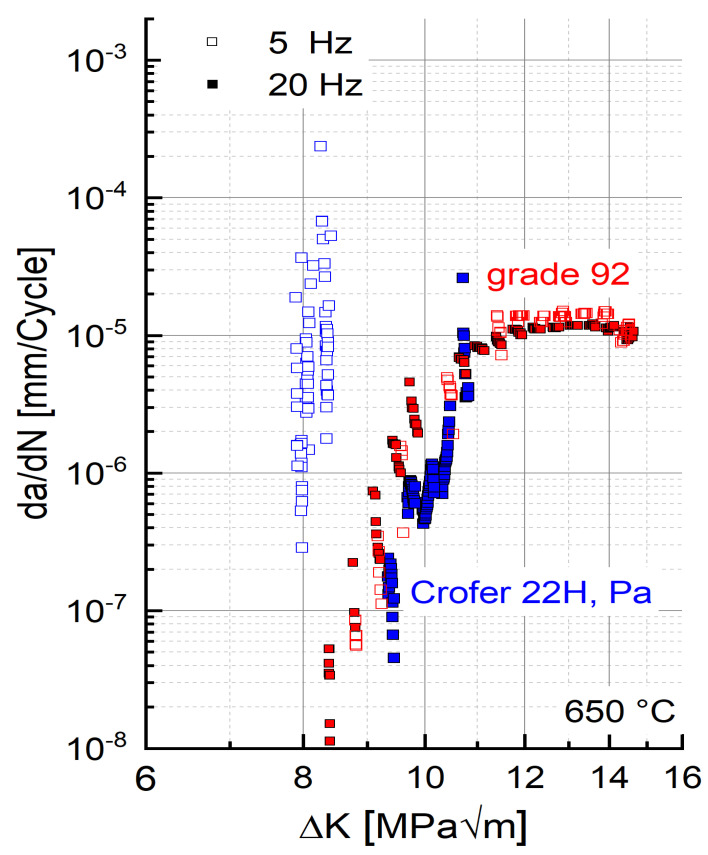
Threshold behaviour of precipitation-annealed Crofer^®^ 22H (PA) and grade 92 at 650 °C.

**Figure 7 materials-15-06280-f007:**
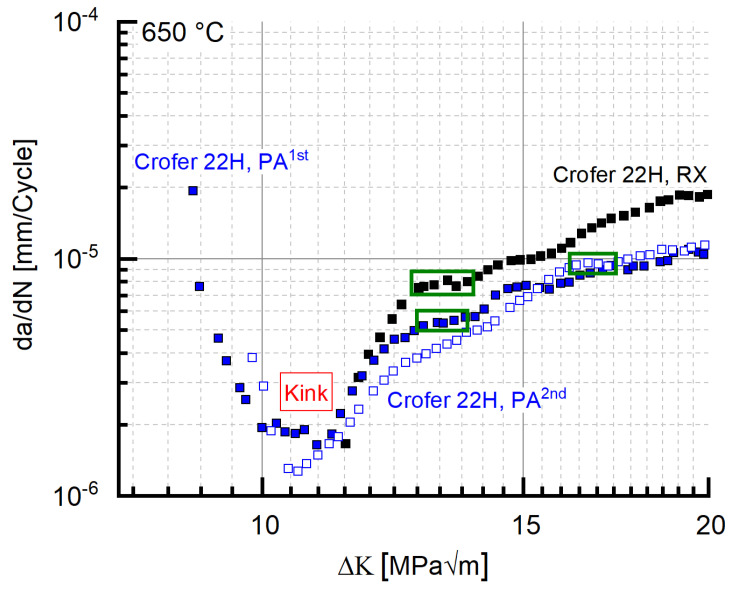
FCG data of Crofer^®^ 22H in the RX and the PA state (f = 20 Hz, T: 650 °C).

**Figure 8 materials-15-06280-f008:**
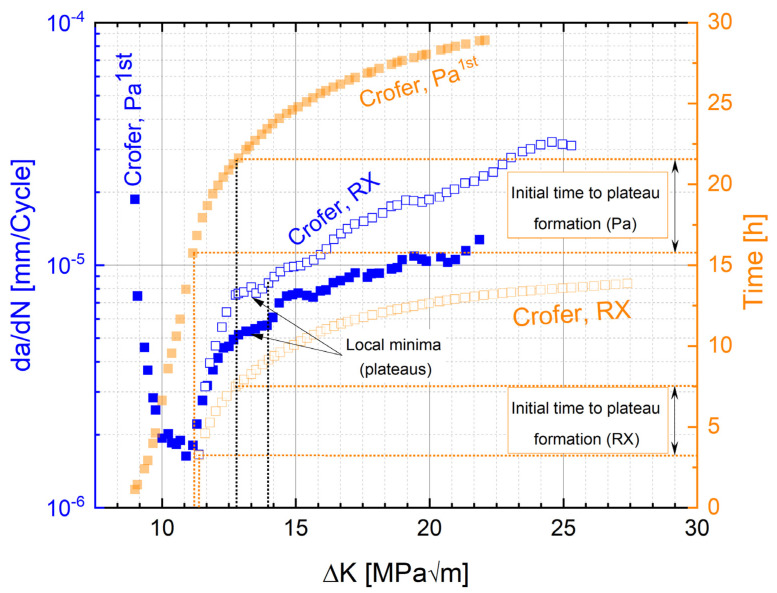
Formation of local minima (plateaus) in Crofer^®^ 22H in da/dN vs. ΔK/ΔK vs. time plot (PA curve: time base corrected by heat treatment duration (+1 h)).

**Figure 9 materials-15-06280-f009:**
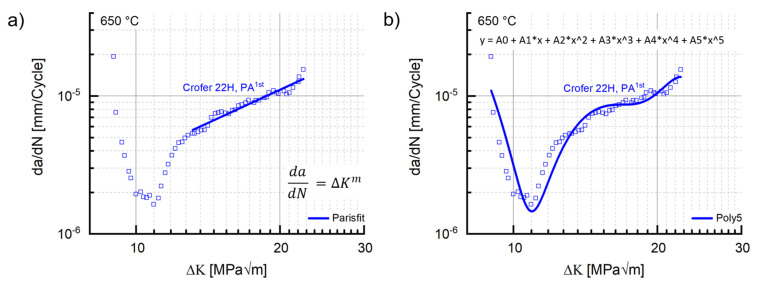
(**a**) Paris and (**b**) fifth-degree polynomial fit to the crack growth curves of precipitation-annealed Crofer^®^ 22H (PA) (f: 20 Hz, T: 650).

**Figure 10 materials-15-06280-f010:**
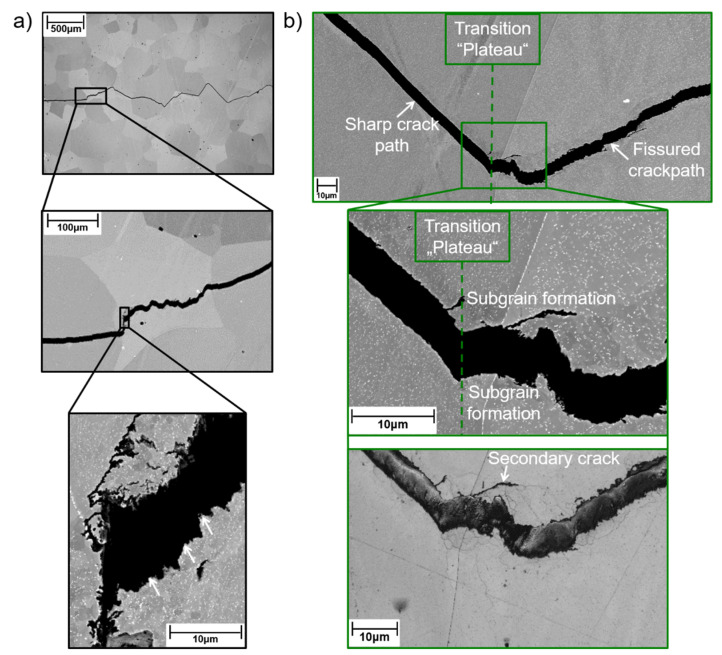
SEM images depict crack path and sub-grain formation (**a**) in the “kink” (da/dN↓/ΔK↑, marked in Figure 7) and (**b**) at the transition from the ΔK↑/da/dN↑ region into one of the “plateaus” (da/dN~constant at ΔK↑; marked by the green boxes in the cyclic crack growth curves in Figure 7).

**Figure 11 materials-15-06280-f011:**
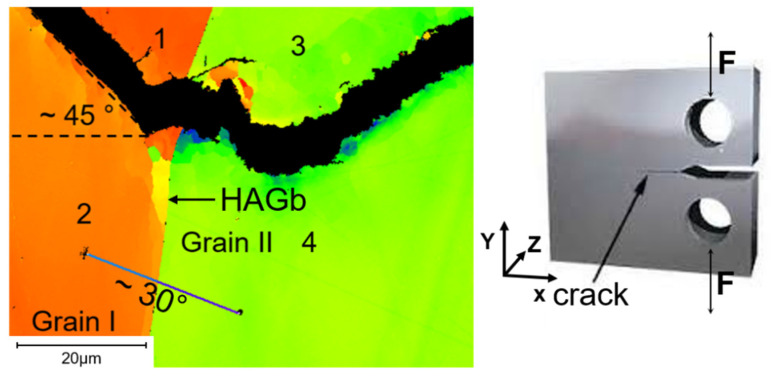
EBSD images show the positions of Schmid factor determination along the fracture path before and within a “plateau” region and corresponding misorientation angles between grains I and II (cf. Figure 7 and Figure 10b).

**Figure 12 materials-15-06280-f012:**
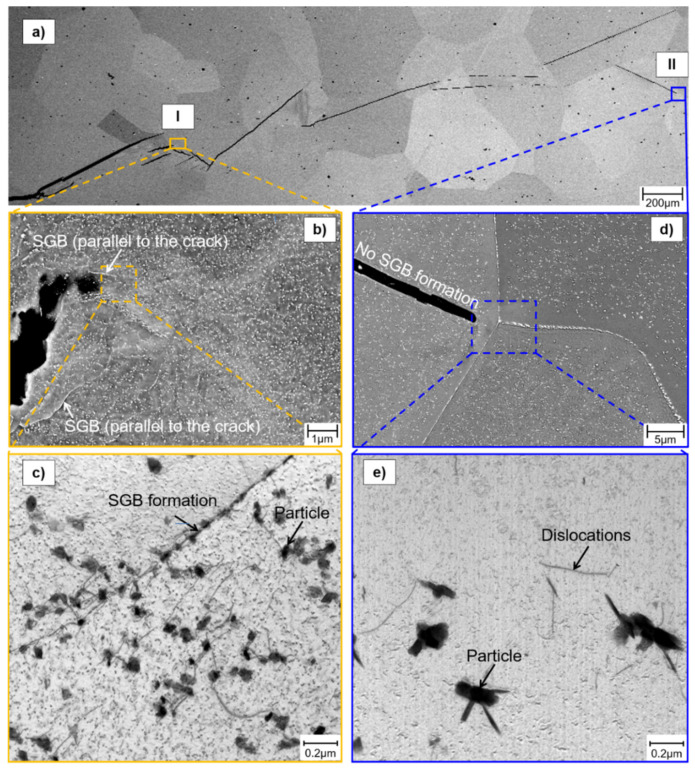
SEM investigations show (**a**) crack path in precipitation-annealed (PA) Crofer^®^ 22H (650 °C, 20 Hz), terminated in the initial range of ΔK↑ with da/dN↓; (**b**,**c**) position I: crack front with sub-grain formation and a multitude of small precipitates; (**d**,**e**) position II: no sub-grain formation, few large precipitates.

**Figure 13 materials-15-06280-f013:**
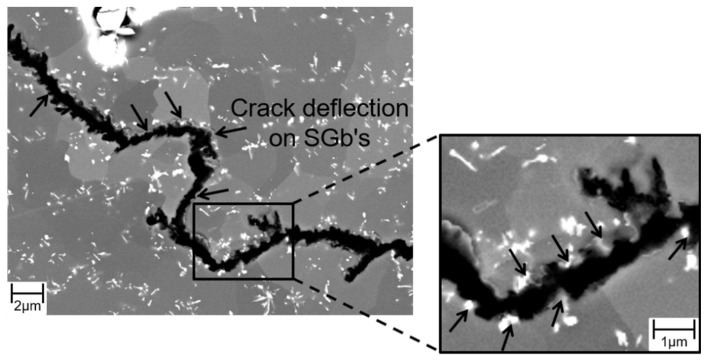
SEM micrographs of crack deflection at Laves phase particle covered sub-grain boundaries in precipitation-annealed (PA) Crofer^®^ 22H.

**Figure 14 materials-15-06280-f014:**
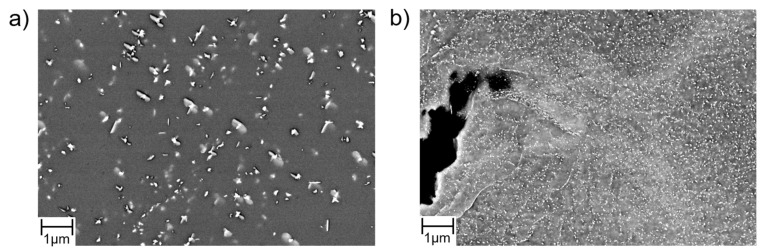
SEM images show a qualitative comparison of Laves phase particle density in (**a**) thermally loaded and (**b**) thermomechanically loaded (PA) Crofer^®^ 22H.

**Figure 15 materials-15-06280-f015:**
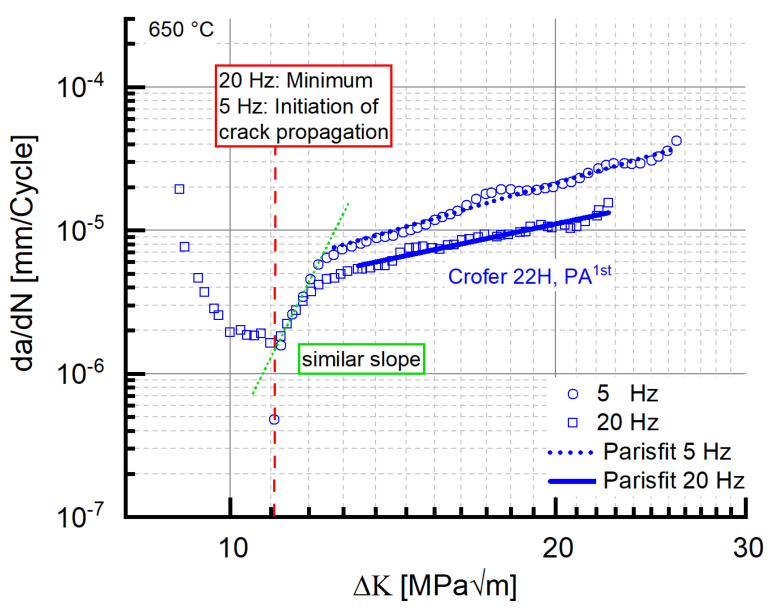
“Frequency impact” on crack propagation rate and corresponding Paris fits in precipitation-annealed Crofer^®^ 22H (PA) at 650 °C.

**Figure 16 materials-15-06280-f016:**
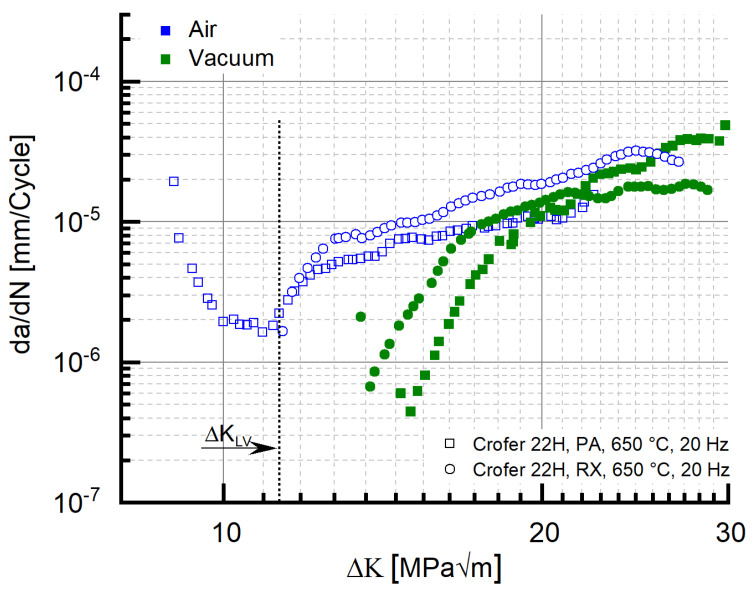
Identification of the limit stress intensity value ΔK_LV_, from which the strengthening mechanisms cannot longer overweigh the increasing stress intensity.

**Figure 17 materials-15-06280-f017:**
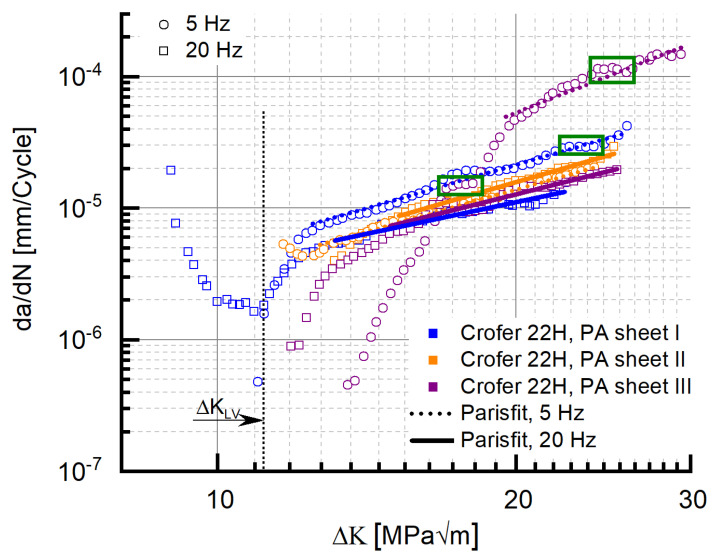
Comparison of the crack propagation behaviour and associated Paris fits of precipitation-annealed Crofer^®^ 22H (PA) specimens taken from three individual sheets from the same alloy heat (f = 20 Hz, T: 650 °C).

**Figure 18 materials-15-06280-f018:**
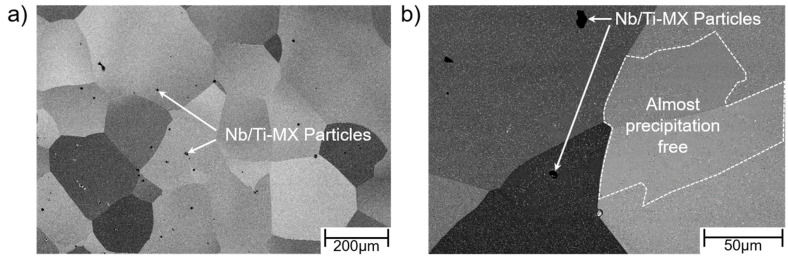
SEM investigations demonstrate (**a**) undesirable Nb/Ti-MX particles in Crofer^®^ 22H (PA) sheet I and (**b**) almost Laves phase particle-free regions within the matrix, which have no additional crack-inhibiting properties except for solid solution strengthening and thus strongly impact the crack propagation behaviour.

**Table 1 materials-15-06280-t001:** Chemical composition of the three Crofer^®^ 22H sheet materials from one melt and the slab of tempered martensite grade 92 (wt. %).

**Crofer^®^22 H**	**C**	**S**	**N**	**Cr**	**Ni**	**Mn**	**Si**	**Mo**	**Ti**
	0.009	0.002	0.02	22.8	0.27	0.45	0.31	0.01	0.06
	**Nb**	**W**	**Cu**	**Fe**	**P**	**Al**	**Mg**	**Co**	**La**
	0.52	2.11	0.02	Balance	0.017	0.015	<0.01	0.01	0.05
**Grade 92**	**C**	**Si**	**Mn**	**P**	**S**	**Cr**	**Mo**	**Ni**	**Cu**
	0.123	0.37	0.49	0.019	0.0010	9.0	0.50	0.34	0.05
	**Nb**	**V**	**Ti**	**Al**	**N2**	**W**	**B**	**Zr**	
	0.075	0.20	0.003	0.015	0.0540	1.69	0.002	0.002	

**Table 2 materials-15-06280-t002:** Test matrix of FCG and threshold experiments (T = 650 °C, R = 0.1).

ID	Sheet nr.	Frequency [Hz]	Atmosphere	Test	Total Number of Tests
Crofer^®^ 22H, RX	I	20	Air	FCG	1
Crofer^®^ 22H, PA	I–III	20	Air	FCG	4
Crofer^®^ 22H, PA, aborted in the range of da/dN↓ with ΔK↑	I	20	Air	FCG	1
Crofer^®^ 22H, PA	I–III	5	Air	FCG	3
Crofer^®^ 22H, RX	I	20	Vacuum	FCG	1
Crofer^®^ 22H, PA	I	20	Vacuum	FCG	1
Crofer^®^ 22H, PA	I	20	Air	ΔK_th._	1
Crofer^®^ 22H, PA	I	5	Air	ΔK_th._	1
grade 92	-	20	Air	ΔK_th._	1
grade 92	-	5	Air	ΔK_th._	1

**Table 3 materials-15-06280-t003:** Frequency dependency of the threshold values (ΔK_th._) of Crofer 22H (PA) and grade 92 (650 °C).

ID	Frequency [Hz]	ΔK_th._ [MPa√m]
Crofer^®^ 22H	5/20	8/9.4
grade 92	5/20	8.8/8.4

**Table 4 materials-15-06280-t004:** Nominal fatigue crack growth parameters (Paris fit) in dependence of annealing state (f = 20 Hz, T: 650 °C).

ID	Coefficient C	Exponent m
Crofer^®^ 22H, RX	5.87 × 10^−8^	1.9
Crofer^®^ 22H, PA^1st^	9.30 × 10^−8^	1.6
Crofer^®^ 22H, PA^2nd^	8.78 × 10^−8^	1.6

**Table 5 materials-15-06280-t005:** Schmid factors (F||Y) of the active slip systems from positions 1–4, illustrated in Figure 11.

Position	Schmid Factor $\left\{110\right\}\;\langle \bar{1}11\rangle$	Schmid Factor $\left\{112\right\}\;\langle 11\bar{1}\rangle$	Schmid Factor $\left\{123\right\}\;\langle 11\bar{1}\rangle$
1 (Grain I)	0.48	0.48	0.50
2 (Grain I)	0.47	0.47	0.49
3 (Grain II)	0.41	0.43	0.44
4 (Grain II)	0.41	0.44	0.44

**Table 6 materials-15-06280-t006:** “Frequency dependent” fatigue crack growth parameters in precipitation-annealed Crofer^®^ 22H (PA) at 650 °C.

ID	Coefficient C	Exponent m
Crofer^®^ 22H (PA), 20 Hz	9.30 × 10^−8^	1.6
Crofer^®^ 22H (PA), 5 Hz	3.04 × 10^−8^	2.2

**Table 7 materials-15-06280-t007:** Summary of frequency-dependent fatigue crack growth parameters of different Crofer^®^ 22H (PA) sheets from the same alloy heat (650 °C).

ID	Coefficient C	Exponent m
Crofer^®^ 22H, Pa^1st^, 20 Hz, sheet I	9.30 × 10^−8^	1.6
Crofer^®^ 22H, Pa, 5 Hz, sheet I	3.04 × 10^−8^	2.2
Crofer^®^ 22H, Pa, 20 Hz, sheet II	1.69 × 10^−8^	2.3
Crofer^®^ 22H, Pa, 5 Hz, sheet II	2.30 × 10^−8^	2.2
Crofer^®^ 22H, Pa, 20 Hz, sheet III	4.41 × 10^−8^	1.9
Crofer^®^ 22H, Pa, 5 Hz, sheet III	7.24 × 10^−9^	3

## Data Availability

Not applicable.

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
