# Peer review of "Active Crack Obstruction Mechanisms in Crofer® 22H at 650 °C"

_materials, 2022, doi:10.3390/ma15186280_

Round 1
Reviewer 1 Report
Dear Editor
The work has been very well planned and executed, and a lot of experiments have been considered. The numerical approaches, tracing crack initiation/propagation behavior and correlation with microstructural evolution has been well explained. I recommended this paper for publication. Please consider the following comments before publication:
Comment 1: The schematic diagram of TMP cycle (with all details) should be added in experimental section, and brief explanation regarding how to control the correlation of compressed air force and rate of cooling.
Comment 2: All of the conducted TMP tests is laid in low cycle regime, so it is expected that the author mainly focuses on crack propagation mechanisms compared with crack initiating one.
Sincerely yours
Reviewer 2 Report
The manuscript entitled: "Active Crack Obstruction Mechanisms in Crofer® 22H at 650 °C" compares thermomechanical fatigue resistance and long crack propagation of the advanced ferritic-martensitic steel grade 92 and Crofer® 22H. The manuscript is well written. Therefore, I suggest publications after minor corrections. Here's some specific comments:
1. In the 2.1 Experimental materials, the authors should briefly describe the method of preparation or the process of obtaining the sample. So, I suggest moving the other details to the introduction section.
2. The axis title, units and text shown in some figures are quite large compared to the font size in the text. The authors should adjust them to a proper size.
3. The authors should add the header to the Table 1 and adjust the headers of Table 3 to match (and clear) with the data.
4. The authors should specify the technique used to investigate microstructure in the figure caption.
